# Immunogenicity of BNT162b2 Vaccine in Patients with Inflammatory Bowel Disease on Infliximab Combination Therapy: A Multicenter Prospective Study

**DOI:** 10.3390/jcm10225362

**Published:** 2021-11-18

**Authors:** Mohammad Shehab, Mohamed Abu-Farha, Fatema Alrashed, Ahmad Alfadhli, Khazna Alotaibi, Abdullah Alsahli, Thangavel Alphonse Thanaraj, Arshad Channanath, Hamad Ali, Jehad Abubaker, Fahd Almulla

**Affiliations:** 1Division of Gastroenterology, Department of Internal Medicine, Mubarak Alkabeer University Hospital, Kuwait University, Kuwait City 47060, Kuwait; ahmadalfadhli@hotmail.com; 2Department of Biochemistry and Molecular Biology, Dasman Diabetes Institute (DDI), Dasman 15462, Kuwait; mohamed.abufarha@dasmaninstitute.org (M.A.-F.); hamad.ali@dasmaninstitute.org (H.A.); jehad.abubakr@dasmaninstitute.org (J.A.); 3Department of Pharmacy Practice, Faculty of Pharmacy, Health Sciences Center (HSC), Kuwait University, Jabriya 13110, Kuwait; Alrashed.ffre@gmail.com; 4Department of Internal Medicine, Adan Hospital, Ministry of Health, Hadiya 46969, Kuwait; Khaznaalotaibi@outlook.com; 5School of Medicine, University of Jordan, Aman 2V5F, Jordan; Abdullah12sahli@gmail.com; 6Department of Genetics and Bioinformatics, Dasman Diabetes Institute (DDI), Dasman 15462, Kuwait; alphonse.thangavel@dasmaninstitute.org (T.A.T.); arshad.channanath@dasmaninstitute.org (A.C.); 7Department of Medical Laboratory Sciences, Faculty of Allied Health Sciences, Health Sciences Center (HSC), Kuwait University, Jabriya 13110, Kuwait

**Keywords:** immunogenicity, IBD, infliximab, azathioprine, vaccine, BNT162b2, COVID-19

## Abstract

Background: Vaccination is a promising strategy to protect vulnerable groups like inflammatory bowel disease (IBD) patients against COVID-19 and associated severe outcomes. COVID-19 vaccine clinical trials excluded IBD patients taking infliximab with azathioprine or 6-mercaptopurine (infliximab combination). Therefore, we sought to evaluate serologic responses to COVID-19 vaccination with the mRNA vaccine, BNT162b2, in patients with IBD receiving infliximab combination therapy compared with healthy participants. Method: This was a multicenter prospective study. Patients with IBD were recruited at the time of attendance at infusion center between 1 August 2021, and 15 September 2021. Our primary outcome were the concentrations of SARS-CoV-2 antibodies 4–10 weeks after vaccination with two doses of BNT162b2 vaccine in patients with IBD taking infliximab combination therapy (study group) compared with a healthy participants group (control group). Both study and control groups were matched for age, sex, and time-since-last-vaccine-dose using optimal pair-matching method. Results: In total, 116 participants were recruited in the study, 58 patients in the study group and 58 in the control group. Median (IQR) IgG concentrations were lower in the study group (99 BAU/mL (40, 177)) than the control group (139 BAU/mL (120, 188)) following vaccination (*p* = 0.0032). Neutralizing antibodies were also lower in the study group compared with the control group (64% (23, 94) vs. 91% (85, 94), *p* < 0.001). The median IgA levels in the study group were also significantly lower when compared with the control group (6 U/mL (3, 34) vs. 13 U/mL (7, 30), *p* = 0.0097). In the study group, the percentages of patients who achieved positive IgG, neutralizing antibody and IgA levels were 81%, 75%, and 40%, respectively. In the control group, all participants (100%) had positive IgG and neutralizing antibody levels while 62% had positive IgA levels. Conclusion: In patients with IBD receiving infliximab combination therapy, SARS-CoV2 IgG, IgA, and neutralizing antibody levels after BNT162b2 vaccination were lower compared with healthy participants. However, most patients treated with infliximab combination therapy seroconverted after two doses of the vaccine.

## 1. Introduction

Inflammatory bowel disease (IBD) is an immune-mediated chronic inflammatory disease. The health of patients with IBD during the coronavirus disease 2019 (COVID-19) pandemic has been an area of concern due to increased susceptibility to infections [1,2]. As with other immune-mediated inflammatory diseases, patients with IBD may require immunosuppressive drugs such as corticosteroids, thiopurines, tumor necrosis factor inhibitors (anti-TNFs), integrin receptor antagonists, anti-IL/12/23 inhibitors, and Janus kinase (JAK) inhibitors to achieve and maintain disease response and remission. The use of such medications has raised concerns regarding the increased susceptibility to acute respiratory syndrome coronavirus 2 (SARS-CoV-2) infection. Indeed, recent studies have confirmed the association of corticosteroids and severe COVID-19 outcomes such as hospitalization and death [3,4,5,6]. Therefore, expert consensus advocates that patients with IBD should be vaccinated against SARS-CoV-2 [7,8].

The goal of vaccination is to produce a long-term immunity against infection, and this can be achieved, in part, by production of pathogen-specific antibody. Immunosuppressive drugs may reduce the effectiveness of some vaccines. Studies showed that response to pneumococcal [9], influenza [10], and hepatitis A [11] and B [12] vaccination in patients with IBD receiving immunosuppressive agents is diminished compared with that in control individuals. However, the impact of IBD medications on COVID-19 vaccine efficacy is unknown. CLARITY study showed that infliximab is associated with attenuated serological responses to SARS-CoV-2 that were further blunted by immunomodulators used as combination therapy [13]. However, to our knowledge, there is no study examined the impact of infliximab use with azathioprine or 6-mercaptopurine (infliximab combination) on COVID-19 vaccine efficacy compared with healthy individuals. The objective of this study was to evaluate serologic responses to COVID-19 vaccination with the mRNA vaccine, BNT162b2 (Pfizer/BioNTech), in patients with IBD receiving infliximab combination therapy compared with healthy participants.

## 2. Materials and Methods

### 2.1. Recruitment and Eligibility

A prospective multi-center cohort study was conducted at two tertiary care centers. Patients were recruited at the time of attendance at the infusion centers between 1 August 2021, and 15 September 2021. Patients were eligible to be included if they: (1) had confirmed diagnosis of inflammatory bowel disease (IBD) before the start of the study, (2) were receiving infliximab with azathioprine or 6-mercaptopurine for at least 6 weeks or more for induction of remission or with at least one dose of drug received for maintenance of remission in the previous 8 weeks, (3) had received two doses of COVID-19 vaccination with the BNT162b2 vaccine, 3 weeks apart, within 4–10 weeks before recruitment, and (4) were at least 18 years of age or older. Patients were excluded if they only received a single dose of the vaccine or if they were infected or had symptoms of SARS-CoV-2 previously since the start of the pandemic. In addition, patients who received other vaccines than BNT162b2 were excluded. Patients who received corticosteroids two weeks before the first dose of the vaccine up to the time of recruitment were also excluded. Finally, patients taking other immunomodulators such as methotrexate were also excluded. This study was performed and reported in accordance with Strengthening the Reporting of Observational Studies in Epidemiology (STROBE) guidelines [14].

Diagnosis of inflammatory bowel disease (IBD) was made according to the international classification of diseases (ICD-10 version: 2016). Patients were considered to have IBD when they had ICD-10 K50, K50.1, K50.8, K50.9 corresponding to Crohn’s disease (CD) and ICD-10 K51, k51.0, k51.2, k51.3, k51.5, k51.8, k51.9 corresponding to ulcerative colitis (UC) [15].

In the study group, data regarding type and extent of IBD as well as duration of infliximab combination therapy were obtained.

Healthy Participants (control) group were individuals with no history of chronic medical illnesses such as diabetes, hypertension, cardiovascular disease, autoimmune diseases, osteoarthritis, chronic obstructive pulmonary disease, renal disease, asthma, hyperlipidemia, or history of stroke and bleeding disorder. In addition, basic laboratory tests were performed (full blood count, renal function tests, liver function tests, lipid profile, HbA1c, ESR, and CRP) to objectively screen for underlying diseases.

### 2.2. Outcome Measures

Our primary outcome were the concentrations of SARS-CoV-2 antibodies including Immunoglobulin G (IgG), Immunoglobulin A (IgA) and neutralizing antibodies 4–10 weeks after vaccination with two doses of BNT162b2 in patients with IBD receiving infliximab combination therapy (study group) compared with healthy participants (control group).

We performed additional analyses for the study group for possible confounders that can suppress the production of SARS-CoV-2-specific antibodies. The additional analyses were conducted to investigate the effect of age, body mass index (BMI), or time to vaccination, by applying Wilcoxon rank sum test. Age was categorized as above and below 35 years of age. BMI was categorized as below 18.5, 18.5–25, 25–30, and above 30 kg/m^2^. Time to vaccinate was divided into 3 categories: 4–6, 6–8, and 8–10 weeks.

### 2.3. Laboratory Methods

In this study, plasma levels of SARS-CoV-2-specific IgG and IgA antibodies were measured by enzyme-linked immunosorbent assay (ELISA) kit (SERION ELISA agile SARS-CoV-2 IgG and IgA SERION Diagnostics, Wüzburg, Germany) based on the manufacturer’s protocol. Units of IgG levels were reported as binding antibody units (BAU)/mL, and values below 31.5 BAU/mL were considered negative or nonprotective. The IgA levels were reported as Arbitrary Units (AU)/mL, values below 10 AU/mL were considered negative or nonprotective. Neutralizing antibody levels below 20% were considered negative or nonprotective. The positive and negative thresholds were determined as per the manufacturer’s instructions. Results were construed by calculating inhibition rates for samples as per the following equation: Inhibition = (1 − O.D. value of sample/O.D. value of negative control) × 100%.

### 2.4. Statistics

We performed descriptive statistics to characterize the study and control group. Standard descriptive statistics were used to present the demographic characteristics of patients included in this study and their measured antibody levels. Analysis was conducted in R (R Core Team, 2017, Free Software Foundation, Boston, MA, USA). Data are expressed as medians with interquartile range (IQR) unless otherwise indicated. Categorical variables were compared using the Fisher’s exact test or Pearson’s Chi-squared test, and continuous variables were compared with the Kruskal–Wallis rank sum test or Wilcoxon rank sum test. A *p*-value of less than or equal to 0.05 was considered statistically significant.

Both Infliximab combination therapy and healthy participants groups were matched for age, sex, and time-since-last-vaccine-dose using optimal pair-matching method. The technique attempts to choose matches that collectively optimize an overall criterion. The criterion used was the sum of the absolute pair distances in the matched sample. In addition, percentages of positive IgG, neutralizing antibody, and IgA levels were calculated in both groups.

## 3. Results

### 3.1. Baseline Cohort Characteristics

Patients were recruited between 1 August 2021 and 15 September 2021. In total, serology assays to quantify SARS-CoV-2 antibody levels were performed for 116 patients. The number of patients included in both the study group and control group was 58. Mean age was 33 in both groups and body mass index was lower in the study group compared with healthy participants (24.8 vs. 26 kg/m^2^). Most patients in the study group had Crohn’s disease (60%) and 20% were smokers. The mean duration between the serology test and last dose of vaccine was 7 (±2) weeks. The median duration of infliximab combination therapy was 12 months (See Table 1).

### 3.2. Outcomes

Median (IQR) SARS-CoV-2 IgG level was lower in patients treated with infliximab combination therapy (99 BAU/mL (40, 177)) than the healthy participants (139 BAU/mL (120, 188)) following vaccination with BNT162b2 (*p* = 0.0032). Median SARS-CoV-2-neutralizing antibodies were also lower in the study group compared with the control group (64 (23, 94) vs. 91 (85, 94), *p* < 0.001). The median IgA level in the study group was also significantly lower when compared with the control group (6 U/mL (3, 34) vs. 13 U/mL (7, 30), *p* = 0.0097) (see Table 2).

The percentage of patients who achieved positive SARS-CoV-2 IgG levels in the control group was 100%, whereas the percentage of patients with positive SARS-CoV-2 IgG levels (>31.5 BAU/mL) in the study group was 81% (Figure 1). The percentage of participants in the study group with positive SARS-CoV-2-neutralizing antibody level was 100%, and the percentage of patients in the study group was 75% (Figure 2). Finally, the percentage of participants in the control group with positive SARS-CoV-2 IgA antibody levels was 62%, whereas the percentage of patients in the study group was approximately 40% (Figure 3).

Additional analyses were performed in the study group for age, BMI, and time to vaccination. In relation to the median levels of SARS-CoV-2 IgG, neutralizing and IgA antibodies, there was no statistically significant difference found in the age, BMI, and time to vaccination subcategories (see Appendix A).

## 4. Discussion

In this study, we explored the serologic response to two doses of the BNT162b2 vaccine in patients with IBD treated with infliximab combination therapy. We found that IgG, IgA, and neutralizing antibody levels were lower in patients with IBD receiving infliximab combination therapy compared with healthy participants 4–8 weeks following vaccination. However, the majority of patients treated with infliximab combination therapy achieved positive antibody concentrations consistent with those thought to provide protection against SARS-CoV-2 infection.

One study reported diminishing humoral responses within 6 months after receipt of the second dose of BNT162b2 vaccine in a large cohort of health care workers [16]. However, another study observed a rapid decay in anti-SARS-CoV-2 antibodies, as early as 14 to 18 weeks, after completing the vaccination in patients with IBD receiving infliximab [17]. Thus, in our study, we only included patients who completed their second dose of vaccine within 4–8 weeks of recruitment.

One cohort study [18] conducted in two centers in the United States and which included 133 patients with inflammatory diseases and 53 immunocompetent patients found that 88.7% of patients with inflammatory disease achieved seroconversion after two doses of the BNT162b2 or mRNA-1273 vaccine. Similar to our study, the authors found that the anti-SARS-CoV-2 antibody levels in patients with inflammatory disease were lower compared with the immunocompetent patients. They also observed that patients receiving corticosteroids had lower antibody titers after both vaccines. Similarly, in a cohort study of 300 patients, 24 patients received infliximab combination therapy. Of the patients in this group, 88% achieved positive humoral immune response after complete vaccination with an mRNA vaccine [19].

Two recent studies have explored the effect of immunosuppressive medications on serological response to SARS-CoV-2 infection and vaccination. Kennedy et al. found that infliximab is correlated with diminished serological response to infection with SARS-CoV-2 [13]. The study also found that in patients receiving infliximab combination therapy, the serological response to infection was reduced further. Another study [20] reported outcomes in patients with IBD receiving biological therapy after two-dose vaccination of either BNT162b2 or mRNA-1273 vaccine. They found that for all antibodies tested, vedolizumab was associated with lower seropositivity; however, in patients receiving an anti-TNF agent, lower antibody level was only reported for anti-receptor binding domain total Ig. Furthermore, a study conducted by Khan et al. in veteran patients with IBD receiving different immunosuppressive medications found that among the 6578 patients with full vaccination status, no SARS-CoV-2 infections were identified among those taking anti-TNF agents, or ustekinumab [21]. However, they reported that the level of protection in their IBD cohort was lower than that reported in the clinical trials with 80.4% effectiveness vs. >90%. Finally, a recent study [22] investigated whether patients with IBD treated with infliximab have attenuated serological responses to a single dose of BNT162b2 or mRNA-1273 vaccines. They concluded infliximab is associated with lower seropositivity to a single dose of BNT162b2 or mRNA-1273 vaccines in patients with IBD. On the other hand, one study [23] assessed antibody titers in adults with IBD who received mRNA SARS-CoV-2 vaccination and were referred from 18 U.S. gastroenterology practices. Participants were receiving various medication regimens including anti-integrin therapy, anti-interleukin-12/23 therapy, and anti–tumor necrosis factor with or without immunomodulator. They found that 99% of participants had detectable antibodies after 2 weeks regardless of medication regimen. However, SARS-CoV-2-circulating antibodies concentrations decreased across all groups over later time points

Another important finding of our study is that while 75–80% of patients on infliximab combination therapy had positive IgG and neutralizing antibody levels, only 40% had positive IgA levels. SARS-CoV-2-circulating antibodies, particularly the IgG class, have been the major contributor to risk reduction of severe COVID-19 after vaccination [24]. IgG antibodies specifically target the spike protein of SARS-CoV-2, its S1 subunit, or its receptor-binding domain (RBD). This diminishes or completely halt the binding of the virus with the host receptors. Neutralizing antibodies have also been shown to correlate with protection [25]. While most studies [26,27] focused on the role of IgG or neutralizing antibodies in preventing severe COVID-19 illness after vaccination, the potential benefits of IgA antibodies in limiting infection and viral spread have been overlooked [28]. Evidence is now accumulating regarding the IgA serum levels and their correlation to neutralization of SARS-CoV-2 at the mucosal surface [29,30]. One study showed that IgA antibodies predominate the early neutralization of the virus and its appearance following infections seems earlier than that of antibodies of the IgG class [31]. This study also observed a rapid decline in SARS-CoV-2-specific IgA serum levels, thereby bringing into question the long-term protective effect of IgA. This could be a possible explanation of the low level of IgA antibodies in our study as well as in the control group.

Our findings have relevant and immediate clinical implications. This study supports the effectiveness of two dose vaccination in IBD patients receiving infliximab combination therapy. This can increase patient motivation to receive COVID-19 vaccine by demonstration and reassurance of vaccine effectiveness. In addition, we found that the majority of patients on infliximab combination therapy achieved antibody levels that conferred protection; while this prospect requires further study, it is not known whether higher antibody titer thresholds correlate to protection against COVID-19 severe outcomes such as hospitalization and death. Therefore, the British Society of Gastroenterology (BSG) and the Food and Drug Administration (FDA) recommended a third dose (or booster dose) of the SARS-CoV2 vaccine for all patients with IBD receiving immunosuppressive treatment [32,33]. Furthermore, it remains to be seen whether infliximab combination therapy accelerates declining of titers over time, but our results may reassure patients treated with such medications that initial humoral responses to mRNA vaccines are generally robust. Future larger follow up studies are needed.

To our knowledge, this is the first prospective cohort study to compare the serological response of IBD patients on infliximab combination therapy to healthy participants. In addition, our predefined inclusion and exclusion criteria lowers the risk of confounding bias, and patients were equally distributed in terms of demographics characteristics such as age, sex, and BMI. Furthermore, additional analyses were performed within the study group to address the wide range of immunogenicity reported and to rule out any possible effects of confounding factors.

One of the limitations of our study is a small sample size, which may not reflect the precise difference between the two groups. Furthermore, even though we matched our participants in each group for known confounding factors, given the observational design, there is potential for unmeasured factors. For example, we did not have objective markers of active inflammation before immunization, such as ESR, CRP, fecal calprotectin and endoscopic scores. However, given that the median ESR, CRP, and fecal calprotectin at the time of antibodies testing were within normal ranges, our study group participants are unlikely to have active inflammation. In addition, we only assessed positive IgG, neutralizing antibody, and IgA (humoral immunity). However, cellular immunity may also play a role in vaccine efficacy. Finally, we investigated infliximab with azathioprine or 6-mercaptopurine only. Further studies are needed to investigate the effect of other immunomodulators.

## 5. Conclusions

In patients with IBD receiving infliximab combination therapy, SARS-CoV2 IgG, IgA and neutralizing antibody levels were lower compared with healthy participants. However, the majority of patients treated with infliximab combination therapy achieved seropositivity after two doses of the vaccine.

## Figures and Tables

**Figure 1 jcm-10-05362-f001:**
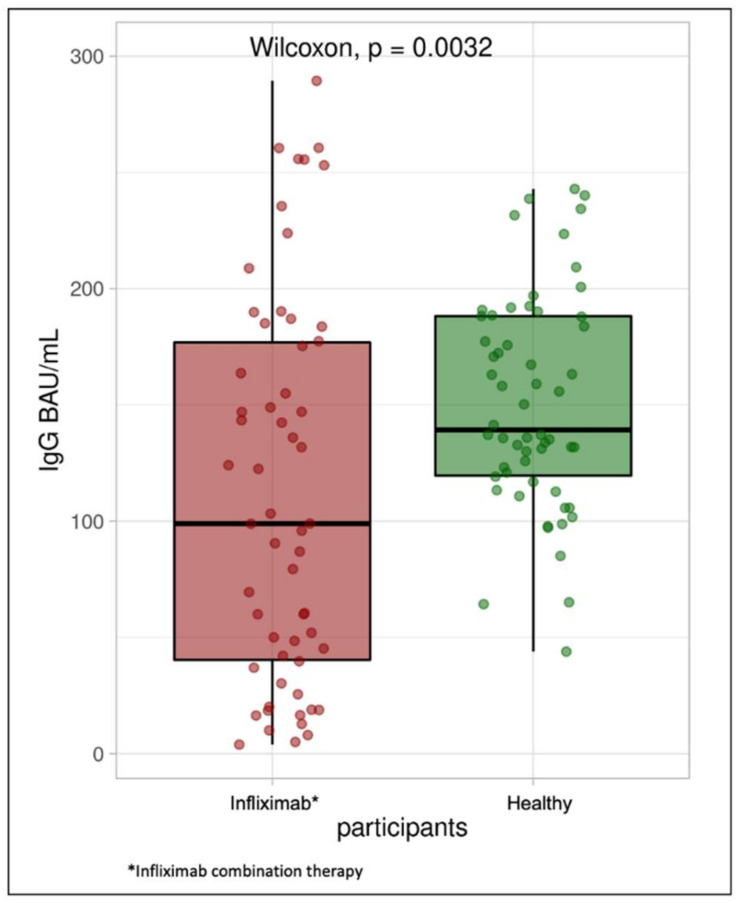
Box plot illustrating SARS-CoV-2 IgG antibody concentrations in the infliximab combination therapy group and healthy participants.

**Figure 2 jcm-10-05362-f002:**
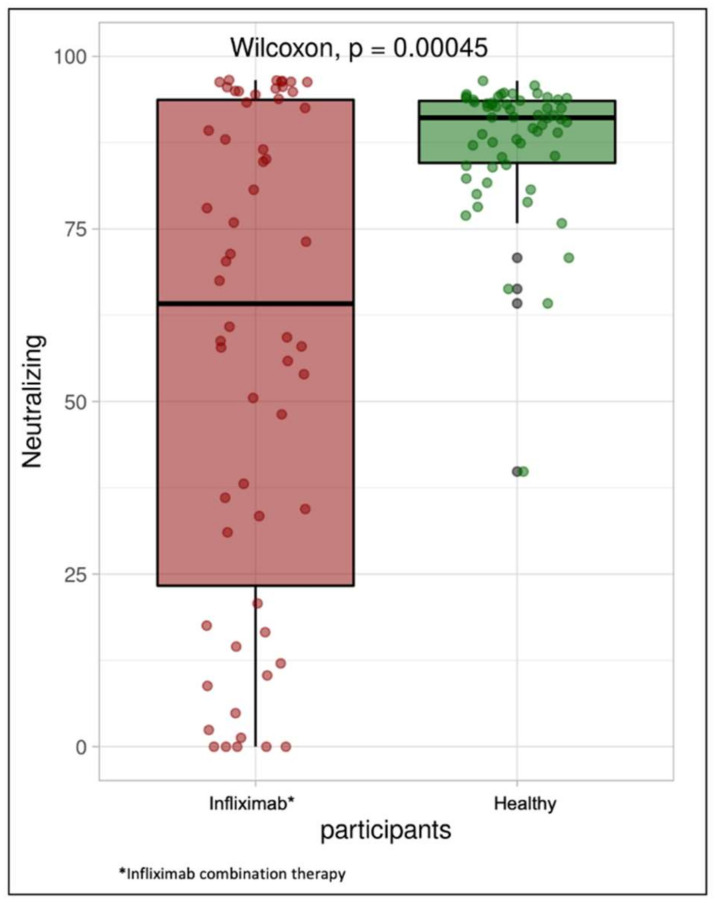
Box plot illustrating SARS-CoV-2-neutralizing antibody concentrations in the infliximab combination therapy group and healthy participants.

**Figure 3 jcm-10-05362-f003:**
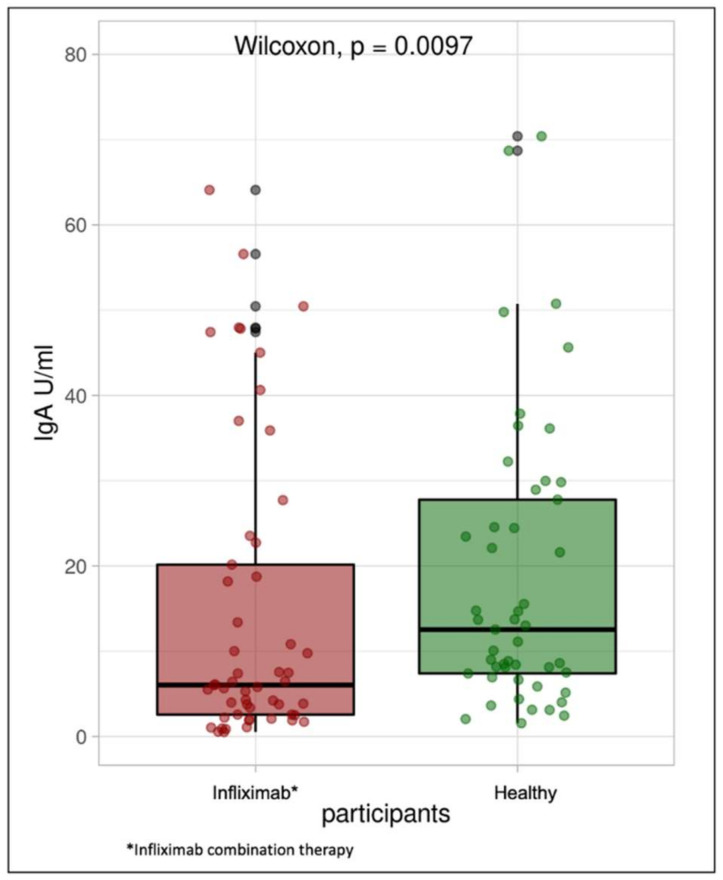
Box plot illustrating SARS-CoV-2 IgA antibody concentrations in the infliximab combination therapy group and healthy participants.

**Table 1 jcm-10-05362-t001:** Baseline characteristics of participants.

Variable	Study Group * (*n* = 58)	Control Group (*n* = 58)
Mean age (years)	33.2	33
Sex *n* (%)		
Male	33 (56%)	31(53%)
Female	25 (44%)	27(47%)
BMI (Median)	24.8	26.0
Smoking *n* (%)	12 (20.8%)	13 (22.4%)
Co-morbidities *n* (%)		
Diabetes	4 (6.8%)	0 (0%)
OSA	1 (1.7%)	0 (0%)
Hypertension	2 (3.4%)	0 (0%)
Cardiovascular Disease	2 (3.4%)	0 (0%)
Arthritis	3 (5.1%)	0 (0%)
Kidney	2 (3.4%)	0 (0%)
Asthma	8 (13.7%)	0 (0%)
Hyperlipidemia	2 (3.4%)	0 (0%)
Duration of infliximab combination therapy (median, months)	12	NA
Disease extent, *n* (%)		
Ulcerative colitis (UC)	24 (40%)	NA
E1: ulcerative proctitis	4 (16.6)	NA
E2: left sided colitis	7 (29.2%)	NA
E3: extensive colitis	13 (54.2%)	NA
Crohn’s disease (CD)	34 (60%)	NA
L1: ileal	17 (50%)	NA
L2: colonic	4 (11.7%)	NA
L3: ileocolonic	12 (35.2%)	NA
L4: upper gastrointestinal	1 (3.1%)	NA
B1: inflammatory	15 (44.1%)	NA
B2: stricturing	9 (26.5%)	NA
B3: penetrating	10 (29.4%)	NA
Lab Parameters		
CRP, mg/L (median)	6.1	5.0
Albumin, g/L (median)	40	40
ESR, mm/h	9	7
Stool fecal calprotectin, ug/g (median)	112	NA

* Infliximab combination therapy.

**Table 2 jcm-10-05362-t002:** Antibody responses: infliximab combination therapy (study) group vs. healthy participants (control) group.

Characteristic	*n*	Overall, *n* = 116 ^1^	Study Group, *n* = 58 ^1^	Control Group, *n* = 58 ^1^	*p*-Value ^2^
IgG BAU/mL	116	134 (86, 184)	99 (40, 177)	139 (120, 188)	0.0032
IgA U/mL	116	8 (4, 24)	6 (3, 20)	13 (7, 28)	0.0097
Neutralizing	116	87 (60, 94)	64 (23, 94)	91 (85, 94)	<0.001

^1^ Median (IQR) or Frequency (%). ^2^ Wilcoxon rank sum test.

## Data Availability

The data presented in this study are available on request from the corresponding author. The data are not publicly available due to local legal and ethical regulations.

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
