# Peer review of "Immunogenicity of BNT162b2 Vaccine in Patients with Inflammatory Bowel Disease on Infliximab Combination Therapy: A Multicenter Prospective Study"

_jcm, 2021, doi:10.3390/jcm10225362_

Round 1

Reviewer 1 Report

The Authors present study titled "Immunogenicity of BNT162b2 Vaccine in Patients with Inflammatory Bowel
Disease on Infliximab Combination Therapy: A Multicenter Prospective Study
Journal: Journal of Clinical Medicine"

overall  this is a fair effort to study impact of COVID 19 immunization in IBD patient population and along lines of similar work published in other journals - including Annals of Internal Medicine (August 2021), while the study lacks novelty - it does help augment our knowledge about immunization in IBD patients. 

  • I feel the manuscript will benefit from careful revision by person proficient in English so that it meets publication standards
  • usually the incorporation of parameters like ESR, CRP, fecal calprotectin, endoscopic IBD scores- are helpful in this population of patients (before/ after immunization) these would have made the study more presentable, still, it is a fair effort.

Thankyou

Author Response

v

overall  this is a fair effort to study impact of COVID 19 immunization in IBD patient population and along lines of similar work published in other journals - including Annals of Internal Medicine (August 2021), while the study lacks novelty - it does help augment our knowledge about immunization in IBD patients. 

  1. I feel the manuscript will benefit from careful revision by person proficient in English so that it meets publication standards

Thank you for all your feedback and comments. Manuscript revised by professional English language editor.

  1. usually the incorporation of parameters like ESR, CRP, fecal calprotectin, endoscopic IBD scores- are helpful in this population of patients (before/ after immunization) these would have made the study more presentable, still, it is a fair effort.

Thank you for these comments. We added ESR, CRP and fecal calprotectin at the time of antibodies testing to table 1. Unfortunately, we do not have tests results before immunization for all patients. However, given that the median ESR, CRP and fecal calprotectin results are within normal limits, our study group are unlikely to have active IBD. We add this point that to the limitation section as well.

“For example, we do not have objective markers of active inflammation before immunization, such as ESR, CRP, fecal calprotectin and endoscopic scores. However, given that the median ESR, CRP, and fecal calprotectin at the time of antibodies testing were within normal, our study group are unlikely to have active inflammation.”

Reviewer 2 Report

I think this research paper with interest. I have two comments. 

  1. I recommend authors will perform additional analysis in IFX group. Is there any factor to suppress the produce of antibody for COVID-19 in that group because it had relatively wide range of immunogenicity?
  2. Did authors check the healthy group by any objective test as well as clinical backgrounds?

Author Response

I think this research paper with interest. I have two comments. 

  1. I recommend authors will perform additional analysis in IFX group. Is there any factor to suppress the produce of antibody for COVID-19 in that group because it had relatively wide range of immunogenicity?

Thank you for this important point. we performed additional analyses for the study group for possible confounders that can suppress the production of SARS- CoV-2-specific antibodies. The additional analyses were conducted to investigate the effect of age, body mass index (BMI), or time to vaccination. Age was categorized as above and below 35 years of age. BMI was categorized as below 18.5, 18.5-25, 25-30 and above 30 kg/m2. Time to vaccinate was divided into 3 categories: 4-6, 6-8, and 8-10 weeks.  Please see methods, results, and additional supplementary tables.

Method:

“we performed additional analyses in the IFX group for possible confounders that can suppress the production of SARS- CoV-2-specific antibodies. The additional analyses were conducted to investigate the effect of age, body mass index (BMI), or time to vaccination, by applying Wilcoxon rank sum test. Age was categorized as above and below 35 years of age. BMI was categorized as below 18.5, 18.5-25, 25-30 and above 30 kg/m2. Time to vaccinate was divided into 3 categories: 4-6, 6-8, and 8-10 weeks.”

Results:

“Additional analyses were performed in the study group for age, BMI, and time to vaccination. In relation to the median levels of SARS-CoV-2 IgG, neutralizing and IgA antibodies, there was no statistically significant difference found in the age, BMI, and time to vaccination subcategories (see supplementary tables s1-3).”

Discussion:

“Furthermore, additional analyses were performed within the study group to address the wide range of immunogenicity reported and to rule out any possible effects of con-founding factors.”

  1. Did authors check the healthy group by any objective test as well as clinical backgrounds?

Thank you, we agree that it is important to clarify that basic laboratory tests were performed (complete blood count, renal function tests, liver function tests, lipid profile, HbA1c, ESR, and CRP). All were within normal limits. Please see method section.

“In addition, basic laboratory tests were performed (full blood count, renal function tests, liver function tests, lipid profile, HbA1c, ESR, and CRP) to objectively screen for underlying diseases

Round 2

Reviewer 2 Report

I have no comments with regard to this revision.